# Comparison of the Stability of a Camu Camu Extract Dried and Encapsulated by Means of High-Throughput Electrospraying Assisted by Pressurized Gas

**DOI:** 10.3390/foods13203280

**Published:** 2024-10-16

**Authors:** Juan David Escobar-García, Cristina Prieto, Emma Talon, Jose M. Lagaron

**Affiliations:** 1R&D Department, Bioinicia S.L., Calle Algepser 65, 46980 Paterna, Spain; etalon@bioinicia.com; 2Novel Materials and Nanotechnology Group, Institute of Agrochemistry and Food Technology (IATA), Spanish Council for Scientific Research (CSIC), Calle Catedrático Agustín Escardino Benlloch 7, 46980 Paterna, Spain; cprieto@iata.csic.es (C.P.); lagaron@iata.csic.es (J.M.L.)

**Keywords:** camu camu extract, room-temperature drying, encapsulation, microparticles, whey protein concentrate, zein, EAPG

## Abstract

This study explores the impact on the stability of drying and the encapsulation of a camu camu extract (CCX) using the non-thermal, high-throughput electrospraying assisted by pressurized gas (EAPG) technique. The dried and encapsulated products by the EAPG processing techniques were compared in terms of total soluble phenolic compounds, antioxidant activity, and storage stability. Whey protein concentrate (WPC) and zein (ZN) were selected as the protective excipients for encapsulation. Dried and encapsulated products were obtained in the form of microparticles, which were smaller and more spherical in the case of the encapsulates. No significant differences were observed in the total polyphenolic content (TSP), and only relatively small differences in the antioxidant capacity were measured among samples. The generated products were subjected to various storage conditions to assess their stability and the preservation of the TSP and the antioxidant properties, i.e., 0% relative humidity (RH) and 4 °C; 0% RH and 21 °C; 23% RH and 21 °C; 56% RH and 21 °C; and UV light exposure. The results indicated that ZN encapsulation notably enhanced the retention of total soluble polyphenols and the antioxidant activity compared to WPC and dried CCX, especially in the ratio of 2:1 (encapsulating polymer: dried CCX). This study demonstrates the potential of protein-based encapsulation, particularly using ZN, for stabilizing bioactive compounds against degradation mechanisms induced by humidity, temperature, or ultraviolet radiation exposure.

## 1. Introduction

There is a growing interest among consumers in natural bioactive compounds that possess the potential to promote health benefits and contribute to the prevention and/or treatment of chronic diseases. Consequently, investigating and researching these natural bioactive compounds is a matter of great importance for numerous laboratories and industries [1,2].

Among the bioactive compounds, phytochemicals stand out due to their ability to inhibit oxidative reactions, thereby protecting the organism from reactive oxygen species, which are implicated in numerous degenerative diseases such as cataracts, atherosclerosis, and cancer [3].

The Amazon region boasts an extraordinary biodiversity of both native and exotic fruits. Camu camu (Myrciaria dubia (H.B.K) McVaugh) is a fruit that is distributed along Bolivia, Brazil, Colombia, Ecuador, Peru, and Venezuela, and it is emerging as a standout due to its remarkable nutritional prospects [4,5]. The composition of camu camu has garnered considerable interest due to its elevated levels of bioactive compounds, such as ascorbic and citric acid and phenolic compounds, including ellagic acid, ellagitannins, and proanthocyanidins [6,7,8], being reported for its purported bioactivities encompassing antihyperglycemic, anti-inflammatory, antihypertensive, and antimicrobial, among other properties [9,10,11]. In this sense, investigations have demonstrated that camu camu exhibits potent antioxidant and anti-inflammatory properties, surpassing those observed in vitamin C tablets when tested in human subjects [12]. Moreover, studies have also reported the potential antiobesity effects of camu camu in a rat model of diet-induced obesity [13]. Additionally, several studies have suggested that camu camu possesses antidiabetic activities, indicating its potential in treating this disease [7,14,15,16].

To develop functional ingredients based on the active properties of camu camu, it is necessary to extract the bioactive compounds from the fruit or the different parts of the plant [10,14], a process that can be followed by a drying step to facilitate its industrial handling. However, it is well-known that some bioactive compounds present low solubility in water or are unstable, making them susceptible to high temperature, light, pH, oxidative stress, and degradative enzymes, which can negatively impact their biological activities, limiting their use in functional foods or nutraceuticals. Proof of that is that Neves et al. observed a significant decrease in the total phenolic content of camu camu fruits after 5 days of cold storage under controlled conditions [17]. Consequently, developing strategies to protect and preserve them, particularly under conditions of processing and storage, and to enhance their bioaccessibility and bioavailability is crucial [18,19]. Among these strategies, encapsulation has emerged as a promising approach to address this challenge.

Encapsulation involves entrapping the bioactive components within a protective matrix, which can slow down degradation processes, prevent the loss of their functionality, and increase bioavailability [2,20]. Therefore, encapsulating bioactive compounds makes extending their shelf life possible, improving their stability throughout various processing stages and final use at consumption and enhancing their compatibility with different formulations [1,21]. Furthermore, the encapsulation strategy could provide controlled release mechanisms and a sustained and targeted delivery of these compounds in the body [22,23,24]. Thus, various encapsulation methods have been explored so far [25]. The choice of the suitable process varies with the active ingredient’s nature, the coating material’s properties, and the final product’s desired characteristics based on the intended use.

Recent evidence underscores that microencapsulation is a potent process to enhance the stability and bioavailability of the naturally occurring bioactive compounds in camu camu, such as ascorbic acid and anthocyanins. Until now, Fujita et al. have reported the microencapsulation of camu camu juice into Arabic gum and maltodextrin via spray drying [15]. Figueiredo et al. reported the microencapsulation of the camu camu extract from pulp and peels by spray drying using maltodextrin, inulin, and oligofructose as encapsulating matrixes [26]. Complementary findings by García-Chacón et al. reveal that spray-drying camu camu pulp with maltodextrin, whey protein, and a 50:50 mixture of both is a strategy to increase the bioactive compounds stability, modulate the fruit sensory properties, and improve their bioavailability [27].

However, among the techniques available, electrohydrodynamic processing, including electrospraying or electrospinning, has emerged as a promising and versatile method for encapsulating bioactive agents. Electrospraying offers several advantages over traditional encapsulation techniques [28]: It enables the production of micro- or nano-scale particles; it is simple and versatile in terms of the materials that can be processed; it allows the production of particles at low or ambient temperatures [28]. However, the main disadvantage of this technology is its low production yield, which has been solved thanks to the integration of electrospraying and gas-driven nebulization [29]. This technology has been named and patented as electrospraying assisted by pressurized gas (EAPG). It involves atomizing the solution through a pneumatic injector using compressed air that nebulizes within a high electric field. This process forms fine droplets that are dried at room temperature and collected as a free-flowing powder. Therefore, this process is suitable for sensitive bioactive compounds. Moreover, it gathers all the advantages of the electrospray process, such as high encapsulation efficiency and control of particle size and morphology, and it is already available on an industrial scale.

Hitherto, camu camu has been encapsulated mainly into polysaccharides [15,27]. However, based on our encapsulation expertise [30,31], food-grade proteins can provide increased stability to sensitive bioactive ingredients [32]. Therefore, this work aimed to study the potential of room-temperature EAPG technology to dry and microencapsulate a camu camu extract (CCX). Whey protein concentrate (WPC) and zein (ZN) were selected as encapsulating agents. The microcapsules were characterized by morphology, total soluble polyphenol (TSP) content, and antioxidant activity. Additionally, this study aimed to assess the storage stability of CCX EAPG-derived microcapsules under different humidity and temperature conditions and against photooxidation.

## 2. Materials and Methods

### 2.1. Materials

Camu camu powder (CCP) was provided by Qomer BioActive Ingredients (Valencia, Spain). Whey protein concentrate (WPC) 80% was purchased from Beurrespa S.L. (Madrid, Spain). According to the distributor, WPC was claimed to contain 81.6% protein (on dry basis) and 7.5% fat. Maize zein (ZN), gallic acid, sodium carbonate, Folin–Ciocalteau reagent, magnesium nitrate, potassium acetate, and silica gel were purchased from Sigma-Aldrich (Saint Louis, MO, USA). Methanol (reagent grade) was purchased from Labbox (Premia de Dalt, Spain). Ethanol 96 vol.% was purchased from Panreac Química SLU (Barcelona, Spain). Deionized water was used throughout this study.

### 2.2. Preparation of Camu Camu Extract (CCX)

The CCP was extracted following a standardized adapted method as described by Fracassetti et al. [8]. A mixture of alcohol and water (85%) was used as the extraction solvent. The extraction process was carried out at a concentration of 5% (*w*/*v*), where 50 g of CCP was mixed with 1000 mL of the extraction solvent. The mixture was homogenized using a magnetic stirrer (H20 series, LBX Instruments, Premia de Dalt, Spain) at room temperature until complete dissolution of CCP. Subsequently, the mixture was sonicated in a SONOPULS HD 2200.2 (Bandelin electronic GMbH and Co. KG, Berlin, Germany) for an additional 30 min. After sonication, the solution was centrifuged at 3000 rpm for 15 min at 4 °C using an Avanti J-26 XPI Beckman Coulter centrifuge (Brea, CA, USA). The resulting supernatant was then filtered under vacuum filtration using Whatman No. 1 filter paper by Cytiva (Marlborough, MA, USA). The filtered supernatant was collected and used for further encapsulation processes.

### 2.3. Preparation of the Polymer Solution

To prepare the encapsulating solutions, a 20% (*w*/*v*) solution of whey protein concentrate (WPC) in water was prepared, and a 4% (*w*/*v*) solution of zein (ZN) in an 85% (*v*/*v*) ethanol aqueous solution. Then, the encapsulating solutions were created by slowly adding the pre-prepared camu camu extract (CCX) to the respective encapsulating matrix solutions at ratios of 1:1 *w*/*w* and 2:1 *w*/*w* (encapsulant to dry weight CCX ratio). The solutions were continuously stirred using a magnetic stirrer (H20 series, LBX Instruments, Premia de Dalt, Spain) at room temperature overnight until a homogenous mixture of polymer solutions was obtained.

### 2.4. EAPG Process

The EAPG process was used to dry the CCX solution and encapsulate the CCX into WPC and ZN, using the Capsultek^TM^ pilot plant from Bioinicia S.L. (Valencia, Spain). The pilot plant consists of a nebulizer that generates aerosol droplets, which are subjected to a high electric field, a drying chamber, and a cyclonic collector. The process was conducted under controlled ambient conditions of 25 °C and 30% relative humidity (RH), as detailed in the study by Busolo et al. [29]. The solution was pumped to the injection unit at a flow rate of 10 mL/min, with an air pressure of 10 L/min and a voltage of 15 kV. The resulting particles were collected from the cyclone as a free-flowing powder, stored in flasks under vacuum at −20 °C, and protected from light to prevent oxidation until further analysis.

The theoretical loading capacity for the CCX encapsulates was 50% for the ratio of 1:1 and 33% for the ratio of 2:1. Furthermore, a 100% encapsulation efficiency was obtained, meaning no significant bioactive loss occurred during the encapsulation process.

### 2.5. Microscopy

Scanning electron microscopy (SEM) analysis of the particle morphology was conducted using a Hitachi S-4800 field-emission scanning electron microscope (Hitachi High Technologies Corp., Tokyo, Japan). The microscope was operated at an electron beam acceleration of 5 kV. Before analysis, approximately 5 mg of the capsules from each sample were coated with a thin layer of gold/palladium to enhance conductivity and imaging quality during SEM observations. The SEM images were captured, and the average diameters of the particles were determined using Image J Launcher v1.41 software developed by the National Institutes of Health (Bethesda, MD, USA).

### 2.6. Color

The color was determined using a chroma meter CR-400 (Konica Minolta, Tokyo, Japan). Color differences between samples were calculated using the CIE Lab* color space, which measures color in three dimensions: lightness (L*), red–green axis (a*), and yellow–blue axis (b*). The total color difference, ΔE*, was determined based on the Euclidean distance between the two points in the Lab space, following the formula:(1)$${\Delta E}^{*}=\sqrt[{2}]{\left[{\left({\Delta L}^{*}\right)}^{2}+{\left({\Delta a}^{*}\right)}^{2}+{\left({\Delta b}^{*}\right)}^{2}\right]}$$

This metric represents the magnitude of the perceptual color difference. According to established thresholds, a ΔE* value below 1 is imperceptible to the human eye, while a value between 1 and 2 can be detected only by experienced observers. A ΔE* value between 2 and 3.5 is noticeable to most observers, values between 3.5 and 5 indicate a clear color difference, and values above 5 indicate that the colors appear distinctly different. This method allows for an objective comparison of color changes, facilitating the assessment of product consistency during storage or processing.

### 2.7. Moisture

The CCX EAPG-derived samples were dried at 40 °C in a vacuum oven (Vaciotem-T, J.P. Selecta, Barcelona, Spain) until reaching a constant weight. Subsequently, the moisture content was determined by measuring the weight differential before and after the drying process.

### 2.8. Assessment of the Stability of CCX Formulations

The stability of CCX formulations was evaluated under different storage conditions of relative humidity (RH) and temperatures and accelerated oxidation conditions under ultraviolet (UV) radiation. To simulate different humidity levels, samples of CCX formulations were placed in desiccators containing either silica gel (0% RH), potassium acetate oversaturated solution (23% RH), or magnesium nitrate oversaturated solution (56% RH). The desiccators were kept at a cooling temperature (4 °C) or ambient temperature (~21 °C) accordingly. The samples were stored under the above conditions and monitored for 40 days. Weekly samples were collected and analyzed.

Regarding UV radiation, a commercially available OSRAM Ultra-Vitalux lamp (OSRAM, Garching, Germany) irradiated the samples with UV light. This lamp consists of a quartz discharge tube and a tungsten filament, producing a blend of radiation like natural sunlight. The lamp was operated at a power of 300 W, and only wavelengths like those present in daylight passed through the special glass bulb. After 1 h of exposure, the radiation between 315–400 nm was measured to be 13.6 W, and the radiation between 280 and 315 nm was measured to be 3 W. Approximately 20 g of capsules were evenly spread on Petri dishes and positioned at 20 cm under the UV lamp. The powder in the Petri dish was stirred daily to maintain a uniform treatment. The thickness of the powder layer was maintained below 5 mm to ensure consistent exposure to UV light. The capsules were subjected to the specified storage conditions, and their stability was evaluated over 40 days. Weekly samples were taken from the capsules for analysis.

The analysis of the samples was specifically made to focus on variations in total soluble polyphenols and antioxidant capacity, and attenuated total reflectance Fourier-transform infrared (ATR-FTIR) spectroscopy was utilized.

### 2.9. Total Soluble Polyphenols

The total soluble polyphenol (TSP) content in the CCX formulations was determined using the Folin–Ciocalteu method described by Singleton and Rossi [33]. For the analysis, 10 mg of the CCX formulations were diluted in ethanol at a concentration of 50% (*v*/*v*). Similarly, an equivalent amount of WPC-CCX capsules was diluted in ethanol at 33% (*v*/*v*), and an equivalent amount of ZN-CCX capsules was diluted in ethanol at 75% (*v*/*v*). The ethanol concentrations utilized in this study were specifically selected to facilitate the optimal solubility of each encapsulating matrix (WPC and ZN) as well as the solubility of the CCX. After dissolution, 20 μL of the capsule’s solution was mixed with 1.2 mL of water and 300 μL of sodium carbonate solution at a concentration of 7.5% (*w*/*v*) in Eppendorf tubes. The mixture was then homogenized for 1 min using a vortex shaker and left to stand for 5 min.

Next, 380 μL of water was added to the mixture, followed by 100 μL of a 10% (*v*/*v*) Folin reagent solution. The reaction proceeded for 15 min under dark conditions and at room temperature. Subsequently, the absorbance of the resulting solution was measured in triplicate at a wavelength of 765 nm using a UV/Vis spectrophotometer (UV4000 Dinko Instruments, Barcelona, Spain). The absorbance values obtained were then used to calculate the total soluble polyphenols content of the samples, expressed in gallic acid equivalents (mg GAE/g of dried CCX).

### 2.10. Antioxidant Activity

The antioxidant capacity of the CCX formulations was evaluated using the DPPH (2,2-diphenyl-1-picrylhydrazyl hydrate) free radical scavenging assay, a widely accepted method for assessing the ability of compounds to neutralize free radicals, which is indicative of their antioxidant activity. CCX-EAPG-derived samples were dissolved in ethanol at specific concentrations, like how TSP determination is performed. Then, they were mixed with a DPPH methanolic solution and followed the DPPH methodology from previous studies [31]. After the incubation period, the absorbance of the resulting mixture was measured at a wavelength of 517 nm using a UV/Vis spectrophotometer (Dinko Instruments, Barcelona, Spain).

### 2.11. Attenuated Total Reflection–Fourier Transform Infrared (ATR-FTIR)

The ATR-FTIR spectra of CCX formulations were measured using a Bruker Tensor 37 FT-IR Spectrometer (Bruker, Ettlingen, Germany) equipped with an ATR sampling accessory called the low-temperature Golden Gate from Specac Ltd. (Orpington, UK). This accessory ensures proper contact between the diamond crystal and the encapsulated samples, enhancing the quality of the spectra. Approximately 50 mg of the capsules were used for each measurement.

The spectra were precisely collected using ATR-FTIR spectroscopy within the 4000–600 cm^−1^ wavenumber range, capturing a comprehensive infrared frequency spectrum crucial for molecular bond characterization. The protocol averaged 10 scans per spectrum to enhance the signal-to-noise ratio, a standard analytical chemistry practice for data quality improvement. Spectral resolution was set at 4 cm^−1^, allowing proper resolution for molecular vibration interpretation.

The OPUS 4.0 data collection software program, developed by Bruker (Ettlingen, Germany), was used to analyze the spectral data. This software enables the processing, analysis, and interpretation of the ATR-FTIR spectra, allowing for the identification and characterization of the encapsulated samples. Measurements were performed in triplicate to ensure the reproducibility and reliability of the results. For comparison purposes, first, the baseline of the analyzed spectra was adjusted, and then the spectra were maximized to the band with the highest intensity in the wavenumber range between 1800 and 800 cm^−1^.

### 2.12. Statistical Analysis

The data were expressed as mean ± standard deviation. To determine the significance of the differences observed among the different samples, an analysis of variance (ANOVA) was conducted. Following ANOVA, a Tukey test was performed to compare the means of multiple groups. In this analysis, differences were considered significant when the *p*-value was less than 0.05. For this analysis, the software used was Statgraphics Centurion Version 17.2.04, developed by Statistical Graphics Corp. (Rockville, MD, USA).

## 3. Results and Discussion

This study investigates the effect of the drying of CCX and its microencapsulation using WPC and ZN via EAPG technology. The resulting microcapsules were analyzed for their morphology, color, total phenolic content, antioxidant activity, and storage stability under different ambient conditions.

### 3.1. Physicochemical Characterization

The obtained particles were characterized in terms of morphology, moisture, and color. Figure 1 provides a comparative analysis of particle morphologies achieved through EAPG technology. When drying neat CCX (Figure 1A), irregular, non-uniform, and agglomerated structures were predominantly observed, resulting in a particle size distribution of 10.01 ± 1.84 μm, as shown in Table 1.

The CCX encapsulates exhibited a spherical morphology, and the average particle sizes are gathered in Table 1. The encapsulates showed an average particle size of around 6 µm with a narrow size distribution and with no significant differences among formulations. These results are consistent with earlier successful EAPG encapsulations of bioactive compounds, such as Eicosapentaenoic acid (EPA) [30] and Dragon’s blood sap (DBS) [31], which produced microparticles with average sizes around 6 µm and 11 µm, respectively.

Figure 2 illustrates the particle size distribution of the CCX EAPG-derived capsules. The samples were monodispersed. Broad size distribution for the neat CCX dried by EAPG and the encapsulates with WPC was observed. However, ZN encapsulates showed smaller particle sizes and a narrower particle size distribution. This difference between WPC and ZN encapsulates can be attributed to the protein composition of the WPC, i.e., 80% WP and 20% lactose, resulting in a broader particle size distribution. Similar observations were made by Prieto et al. when comparing the particle morphology and size distribution when encapsulating algae oil in WP with different protein content [34].

Nevertheless, the particle sizes were consistently below the acceptable detection threshold of 20 µm, ensuring no detectable sensory impact [35], thus preserving the potential for maintaining texture in the final product formulation. This is an advantage of the EAPG process in comparison to conventional technologies. For instance, in a recent study conducted by García-Chacón et al., the use of whey protein concentrate (WPC) and maltodextrin as coating agents were compared during the spray drying process of camu camu. Maltodextrin produced particles of sizes ranging from 1.5 to 12.5 μm, while whey protein resulted in particles ranging from 5 to 45 μm [27]. In another study, Figueredo et al. encapsulated a camu camu extract via spray drying using maltodextrin, inulin, and oligofructose as carrier agents, observing varying particle sizes between 5 and 54 μm [36].

Regarding the moisture content of the samples obtained, it was on average below 5%, as shown in Table 1. This moisture level is widely acknowledged as safe, as it inhibits the growth of most harmful bacteria [37].

Color analysis was conducted on the CCX samples produced through the EAPG technique, and the results are presented in Table 1. The CCX-EAPG sample reflects the natural color of camu camu, which is attributed to its high phenolic content, particularly anthocyanins and carotenoids, along with its notable ascorbic acid content, contributing to its visual appearance [17].

However, the color of the encapsulates resulted in a mixture between the color of the CCX and the color of the pure encapsulating matrices. WPC had L*, a*, and b* values of 57.94 ± 0.18, −0.12 ± 0.07, and 9.52 ± 0.08, respectively, while ZN showed values of 50.14 ± 0.18, 0.93 ± 0.02, and 10.93 ± 0.06, respectively. Thus, encapsulation with WPC caused a noticeable color change, with ΔE* values within the range of 2 to 3.5, meaning that the color difference would be perceptible to most observers but not strikingly different. The WPC encapsulation process slightly lightened the samples and reduced the red hues while enhancing the yellow tones, suggesting that WPC creates a subtle masking effect on the natural pigments without drastically altering the overall color.

Regarding the encapsulation with ZN, the most significant color changes were observed in ZN-CCX 2:1 ratio capsules with ΔE* well above 5. This substantial increase in lightness and reduction in redness suggests a more pronounced influence on the protein color. These results underline the critical role of the matrix composition in influencing the visual appearance of the encapsulates in addition to its protection role.

In addition, it is important to mention that the capsules were readily soluble in water or ethanol depending on the nature of the encapsulating matrix, as no thermal treatment or chemical reaction occurred during the EAPG encapsulation process.

### 3.2. Characterization of Antioxidant Activity and the Polyphenol Content

The antioxidant activity and polyphenol content of the CCX-EAPG-derived capsules are shown in Table 2. Previous studies have demonstrated that camu camu possesses antioxidant capabilities that significantly surpass those of various fruits by over tenfold [27,38,39]. In our investigation, DPPH inhibition percentages for the CCX were 89.06%, which increased to some small extent to 94.07% for CCX-EAPG, potentially attributed to humidity loss during drying by EAPG. Regarding the encapsulates, the 2:1 encapsulation ratio in protein-based formulations was found to be more antioxidant for the 2:1 ratio than for the 1:1 ratio, especially in the case of zein. This observation is consistent with findings from prior studies on dragon blood sap (DBS) with EAPG-delivered capsules [30], where a 1:1 coating-to-core ratio led to decreased DPPH radical scavenging activity. In any case, the differences among samples are relatively small and could be within the experimental error associated with the process and the analytical method. In future studies, other antioxidant tests, such as ORAC, ABTS, or FRAP, will be considered to study the differences between the samples.

To further investigate the stability and antioxidant characteristics of the processed materials, the total soluble polyphenols (TSPs) were also determined using the Folin–Ciocalteu method. The results, shown in Table 2, unveiled consistent TSP levels across all samples, with a narrow range from 1.06 to 1.15 mg GAE/g dried CCX. The consistent retention of total soluble polyphenols (TSPs) indicates the efficiency of the EAPG method in preserving the initial TSP content during drying or encapsulation. Moreover, the comparable TSP values among samples imply that the choice of coating materials did not significantly affect the polyphenol content. These results align with prior research demonstrating the preservation of bioactive compounds, including polyphenols, throughout encapsulation processes [31].

According to the literature, the total phenolic content within camu camu is influenced by factors such as fruit composition, ripeness, and postharvest processing methods. The camu camu flour used in this study was obtained through a comprehensive procedure involving handpicked fruits subjected to fluidized bed drying at temperatures below 60 °C, followed by the fruits being finely ground into a powder and packaged under controlled conditions to safeguard their purity. Regarding the results provided by other authors, Genovese et al. (2008) reported a concentration of 19.92 mg GAE/g in commercial camu camu fruit samples [40]. Similarly, Chirinos et al. (2010) observed varying polyphenolic content among different color stages of camu camu fruit, with concentrations ranging from 12.42 to 15.74 mg GAE/g [41]. Conversely, often overlooked seeds showcased a notably elevated phenolic content of 3.36 mg GAE/g, underscoring their potential as a concentrated source of valuable compounds. Fidelis et al. (2020) extended to seed extracts, documenting values ranging from 6.57 mg GAE/g to 54.20 mg GAE/g [42].

Furthermore, Fracassetti et al. (2013) noted substantial differences in TSP content among various components of camu camu [8]. Specifically, processed derivatives such as flour exhibited a higher concentration of phenolics at 6.73 mg GAE/g, whereas less processed forms like pulp powder contained significantly lower levels at 0.49 mg GAE/g. Despite being rich in phytonutrients, the peel displayed a relatively modest phenolic content of 0.11 mg GAE/g, whereas fresh pulp exhibited 0.09 mg GAE/g. Therefore, this aspect must be taken into account in order to consider the potential of industrialization.

### 3.3. Storage Stability

This study evaluated the stability of EAPG-dried CCX and CCX microencapsulates under various storage conditions. Microcapsules were exposed to diverse relative humidity (0%, 23%, and 56%) and temperatures (4 °C and 21 °C) over 40 days. An accelerated aging treatment was conducted to assess photo-oxidation stability by subjecting the microcapsules to UV light irradiation. During this storage time, particle size and morphology revealed no significant changes in the obtained samples. TSPs and DPPH were used to quantify the stability of the bioactive, whereas ATR-FTIR was used to provide a qualitative description of the chemical changes that occurred under the studied storing conditions.

#### 3.3.1. Polyphenol Content Stability

Throughout the study, TSP evolution was monitored for each formulation to indicate its storage stability. The EAPG-dried CCX exhibited a time-dependent reduction in TSPs throughout the 40-day storage period (Figure 3A). Significant differences (*p* < 0.05) were observed among the storage conditions. Microcapsules stored under low humidity and temperature (0% RH, 4 °C) displayed the lowest TSP reduction. Conversely, exposure to UV light irradiation or higher relative humidity (23% and 56% RH) and moderate temperature (21 °C) substantially reduced TSP. This observation is consistent with prior scientific investigations emphasizing the vulnerability of polyphenols to temperature [43] and photodegradation when exposed to UV light [44] and humidity [45,46].

Regarding the encapsulates, the WPC-CCX 1:1 ratio microparticles (Figure 3B) displayed significant TSP decay across all tested storage conditions throughout the 40 days, although to a lesser extent than EAPG-dried CCX. Conversely, WPC-CCX microparticles at a 2:1 ratio (Figure 3C) demonstrated substantial TSP retention under most conditions, except for the high humidity (56% RH) experiment, where significant differences in TSP decay occurred.

In contrast, ZN-CCX microparticles at both 1:1 and 2:1 ratios (Figure 3D,E, respectively) exhibited superior TSP retention under most storage conditions, highlighting the efficiency of ZN in polyphenol protection. However, ZN encapsulation at a 1:1 ratio was seen to be less effective in TSP protection under high humidity, as evidenced by a significant decrease in TSP content at 56% RH. Therefore, the encapsulation with ZN at a 2:1 ratio exhibited the best preservation even under high relative humidity and UV light.

#### 3.3.2. IP-DPPH Storage Stability

This study investigated the stability of the antioxidant activity measured as DPPH inhibition in the EAPG-dried CCX and CCX microencapsulates under the studied storage conditions, including exposure to UV light as an accelerated photooxidation treatment (Figure 4).

Under low humidity conditions (0% RH), regardless of temperature, the dried CCX (Figure 4A) showed less than 20% decay in DPPH inhibition over the 40 days. Conversely, exposure to higher humidity levels (23% and 56%) or UV light resulted in a notable decrease in DPPH inhibition, up to approximately 30%. The effect of the high relative humidity and UV light on the stability of the antioxidant activity is consistent with the observations reported for dragon blood sap [45] and curcumin [47].

Regarding the formulations of the encapsulates, it was found that the same conditions affected the stability of the antioxidant activity. However, the encapsulant materials helped to decrease the observed decay. The microcapsules with a higher encapsulating material ratio (2:1) offered better protection against degradation compared to those with a lower ratio (1:1). WPC-CCX microcapsules displayed less DPPH inhibition compared to ZN-CCX microcapsules, suggesting that ZN provides better protection against the degradation of antioxidant activity. This is consistent with the stability results for TSPs and with a previous work analyzing the stability of dragon’s blood sap against different storage conditions [31].

### 3.4. ATR-FTIR Analysis

Attenuated total reflection–Fourier transform infrared spectroscopy (ATR-FTIR) analysis of the EAPG-dried CCX is shown in Figure 5. The band envelope at ca. 3260 cm^−1^ is primarily associated with O-H stretching vibrations, a chemical group present in the main bioactive compounds of camu camu such as citric acid, ascorbic acid, or phenolic compounds [44,45], but that may also be associated with residual moisture or the residual ethanol from the extraction process. The band at ca. 2920 cm^−1^ is attributed to alkyl -C-H stretching vibrations [48]. The band at around 1675 cm^−1^ is assigned to the carbonyl group [49], a common chemical group in the bioactive compounds prevalent in camu camu [4,6]. The band at ca. 1600 cm^−1^ is associated with alkenyl groups also characteristic of the main bioactive molecules present in camu camu [50,51]. The spectrum also features a band at ca. 1440 cm^−1^ linked to -CH deformation and aromatic ring vibrations [52], also characteristics of some of the bioactive molecules of camu camu such as flavonoids and proanthocyanidins [27,39]. The band at around 1338 cm^−1^ is related to C-C bending vibrations, while the band envelope from 1300 to 1000 cm^−1^ is associated with C-O stretching vibrations related to ether, esters, and alcohol groups, even aromatic alcohols also present in the characteristic bioactive compounds of camu camu such as ascorbic acid, flavonoids, and proanthocyanidins [53]. Lastly, out-of-plane C-H vibrations from 900 to 700 cm^−1^ provide insights into the aliphatic hydrocarbon composition of the CCX [26,49].

The main characteristic bands of the camu camu dried extract were defined, and ATR-FTIR was used to perform a qualitative analysis to determine the storage stability of the different formulations. Figure 6 illustrates the ATR-FTIR spectral evolution of the EAPG-dried CCX microparticles after 40 days under the studied storage conditions. Significant changes in the spectrum were primarily observed in the 1800–800 cm^−1^ wavenumber range, which could indicate CCX degradation. The spectra were maximized to their highest intensity for clearer comparisons to compare relative changes among bands. After a 40-day storage period, the most significant changes in the dried CCX were observed when samples were subjected to high humidity conditions and exposure to UV light (as depicted in Figure 6E,F). These were mainly detected in the band envelopes at ca. 1800–1600 cm^−1^ and at ca. 1200–900 cm^−1^. The first band envelope is ascribed to the carbonyl stretching absorption, which showed a decrease in the number of carboxylic acid groups and an increase in the number of ketone and aldehyde groups, which is aligned with the reported reactions of oxidation of the main bioactive compounds in camu camu such as citric acid, ascorbic acid, flavonoids, and proanthocyanidins [54,55,56,57,58]. Regarding the second band envelope at ca. 1200–900 cm^−1^, this is associated with the stretching vibration of the C-O bond. The observed changes in band position and intensity variation were due to the formation of linear and aromatic ether groups due to the oxidation reactions of the bioactive compounds present in the dried CCX [54,55,56,57,58].

Regarding the encapsulates, Figure 7 and Figure 8 provide the evolution of the ATR-FTIR spectra at the different storage conditions for the system WPC-CCX at 1:1 and 2:1 ratios, respectively. In contrast, Figure 9 and Figure 10 represent the spectra of the system ZN-CCX at 1:1 and 2:1 ratios, respectively. Both encapsulant matrices have a protein nature, and consequently, similar characteristic bands were detected; notably, the band located at ca. 1740 cm^−1^ ascribed to the trenching vibration of the C=O bond, the band at ca. 1632 cm^−1^ and at ca. 1520 cm^−1^ ascribed to the bending vibration of the N-H bond of amides I and II, respectively [59,60].

The ATR-FTIR spectroscopy analysis of WPC-CCX encapsulates revealed small shifts in the spectral evolution. The most significant changes were observed when samples were subjected to high humidity (Figure 7E and Figure 8E) and UV light (Figure 7F and Figure 8F), as an intensity variation in the band envelope between 1600 and 1800 cm^−1^ as a consequence of the formation of aldehydes, ketones, and esters as oxidation products of the bioactive compounds of camu camu [54,55,56,57,58] and a small decrease in the band associated with the secondary amine at ca 1520 cm^−1^, probably due to a slight degradation of the protein. The band envelope between 1200 and 900 cm^−1^ did not show significant differences, probably because the changes in the spectra might be hidden by the bands of the protein. No significant differences were detected for the ratios studied.

The spectral evolution of ATR-FTIR for ZN-CCX encapsulates at 1:1 and 2:1 encapsulation ratios is shown in Figure 9 and Figure 10, respectively. After 40 days of experimentation, slight changes become discernible at high relative humidity and under UV light radiation. Main variations were detected in the 1800–1600 cm^−1^ band envelope as a consequence of the increase in aldehyde and ketone as ester groups due to the progression of the oxidation reactions [54,55,56,57,58] and a slight decrease in intensity in the band related to the amide II of the protein. The band envelope between 1200 and 900 cm^−1^ did not show significant differences, probably because the changes in the spectra might be hidden by the bands of the protein. The ZN-CCX 2:1 encapsulation ratio (Figure 10) exhibits fewer changes than the 1:1 encapsulates, results that agree with the observations made in previous sections.

## 4. Conclusions

This study demonstrates for the first time the impact on the stability of the electrohydrodynamic EAPG technique to dry or encapsulate camu camu extract (CCX) at n. The resulting microparticles exhibit favorable attributes for application in food products, effectively preserving the bioactive properties of CCX and maintaining consistent levels of total soluble polyphenols (TSPs) and antioxidant activity (DPPH inhibition). The comparative analysis conducted in this study demonstrates the superior effectiveness of the encapsulates, especially of ZN as an encapsulation material compared to WPC, in protecting the bioactive properties of camu camu when subjected to challenging storage conditions. The encapsulation system with a 2:1 ratio also demonstrated a markedly reduced degree of degradation compared to its 1:1 counterpart, particularly in environments with high humidity and UV light exposure. Considering the outcome of this work, it is possible to state that the EAPG technique stands out as a versatile and promising approach for creating small particle size and size distribution powders and encapsulated powders of camu camu. The obtained results show the potential of this technology to positively impact the existing drying and encapsulation procedures, establishing a new standard to guarantee the stability and effectiveness of bioactive compounds across various applications, such as functional foods, cosmetics, agricultural inputs, or nutraceutical products.

## Figures and Tables

**Figure 1 foods-13-03280-f001:**
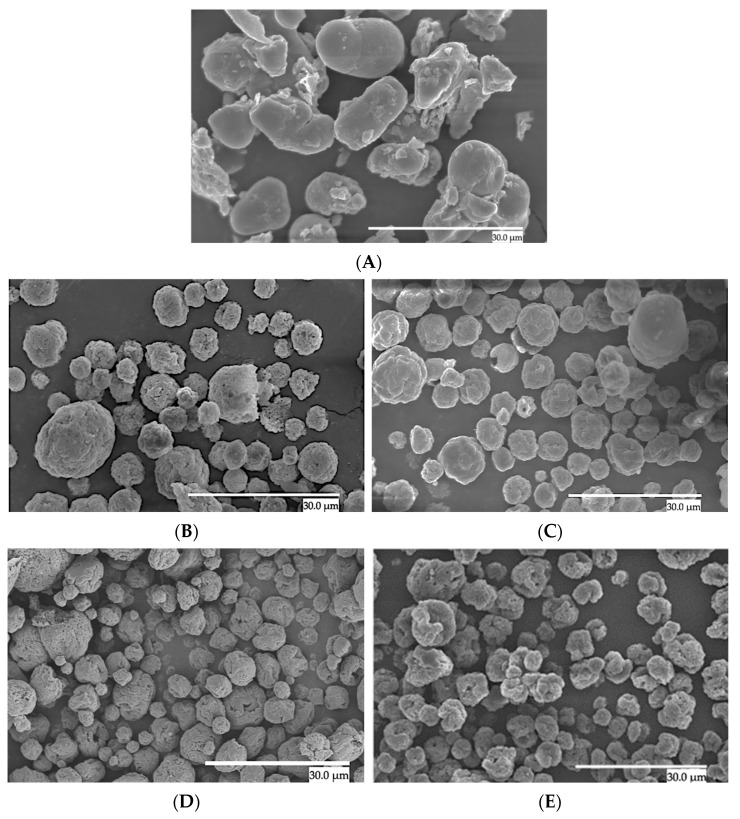
SEM micrographs of dried CCX structures. (**A**) The neat CCX dried using the EAPG process. (**B**) WPC-CCX 1:1 *w*/*w*; (**C**) WPC-CCX 2:1 *w*/*w*; (**D**) ZN-CCX 1:1 *w*/*w*; (**E**) ZN-CCX 2:1 *w*/*w*. The scale bar corresponds to 30 µm.

**Figure 2 foods-13-03280-f002:**
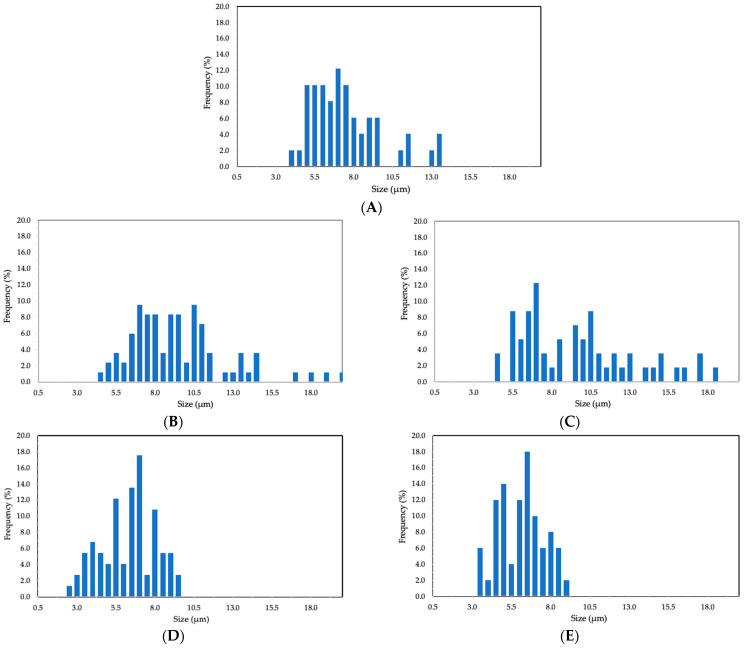
Histograms of dried CCX structures. (**A**) The neat CCX dried using the EAPG process. (**B**) WPC-CCX 1:1 *w*/*w*; (**C**) WPC-CCX 2:1 *w*/*w*; (**D**) ZN-CCX 1:1 *w*/*w*; (**E**) ZN-CCX 2:1 *w*/*w*.

**Figure 3 foods-13-03280-f003:**
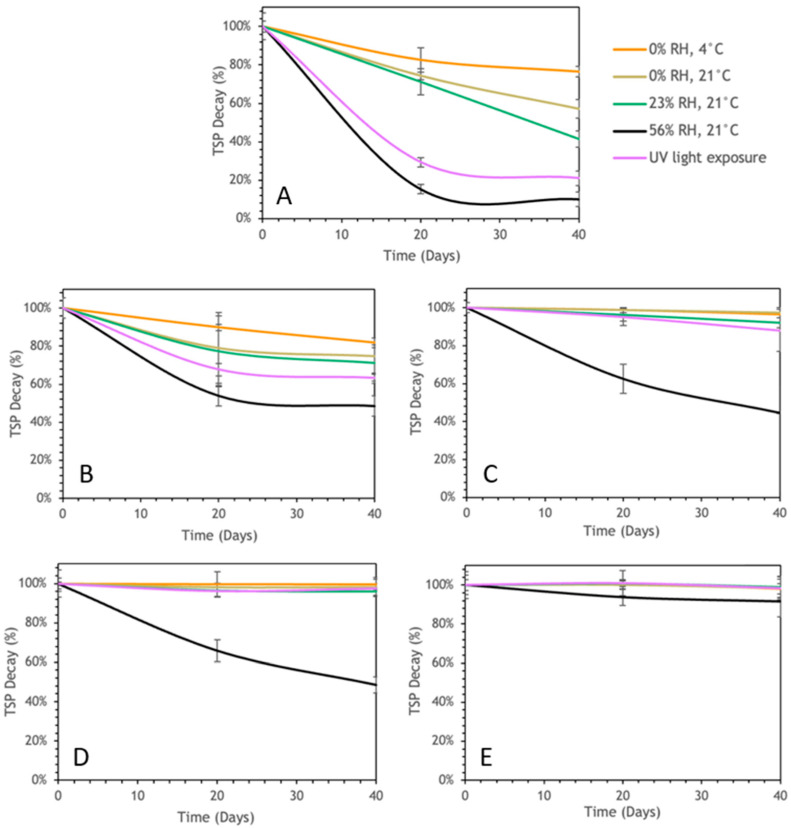
Evolution of the TSP decay of CCX EAPG-derived microcapsules over 40 days under diverse storage conditions. (**A**) EAPG-dried CCX; (**B**) WPC-CCX 1:1 *w*/*w*; (**C**) WPC-CCX 2:1 *w*/*w*; (**D**) ZN-CCX 1:1 *w*/*w*; (**E**) ZN-CCX 2:1 *w*/*w*.

**Figure 4 foods-13-03280-f004:**
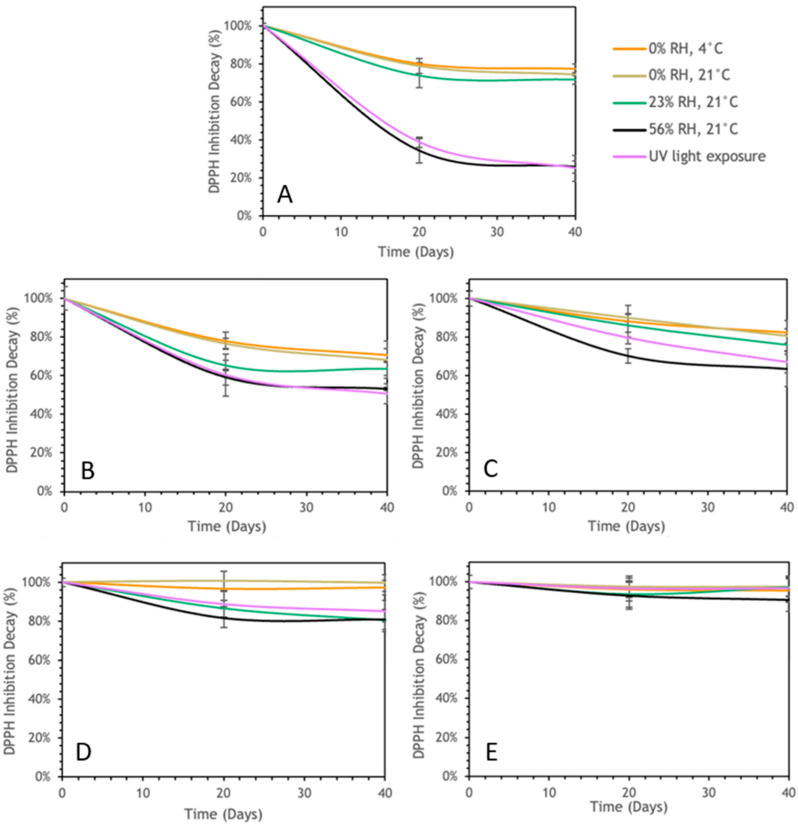
Evolution of the DPPH inhibition decay of CCX EAPG-derived microcapsules over 40 days under diverse storage conditions. (**A**) Non-encapsulated CCX; (**B**) WPC-CCX 1:1 *w*/*w*; (**C**) WPC-CCX 2:1 *w*/*w*; (**D**) ZN-CCX 1:1 *w*/*w*; (**E**) ZN-CCX 2:1 *w*/*w*.

**Figure 5 foods-13-03280-f005:**
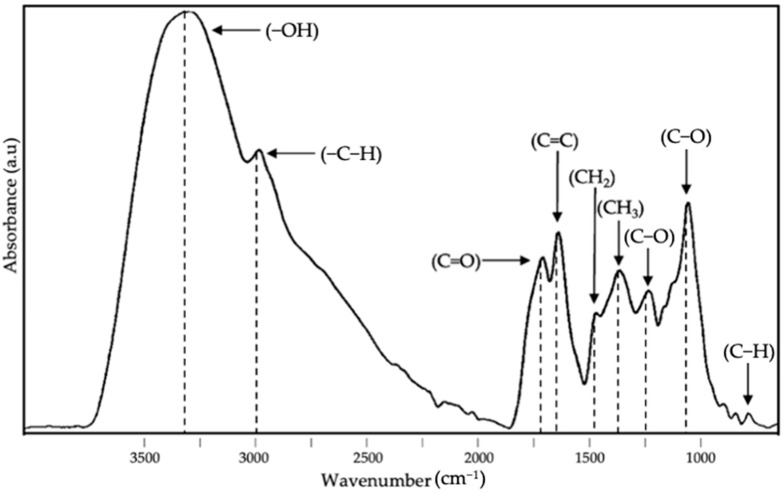
Attenuated total reflection–Fourier transform infrared spectroscopy (ATR-FTIR) of EAPG-dried CCX.

**Figure 6 foods-13-03280-f006:**
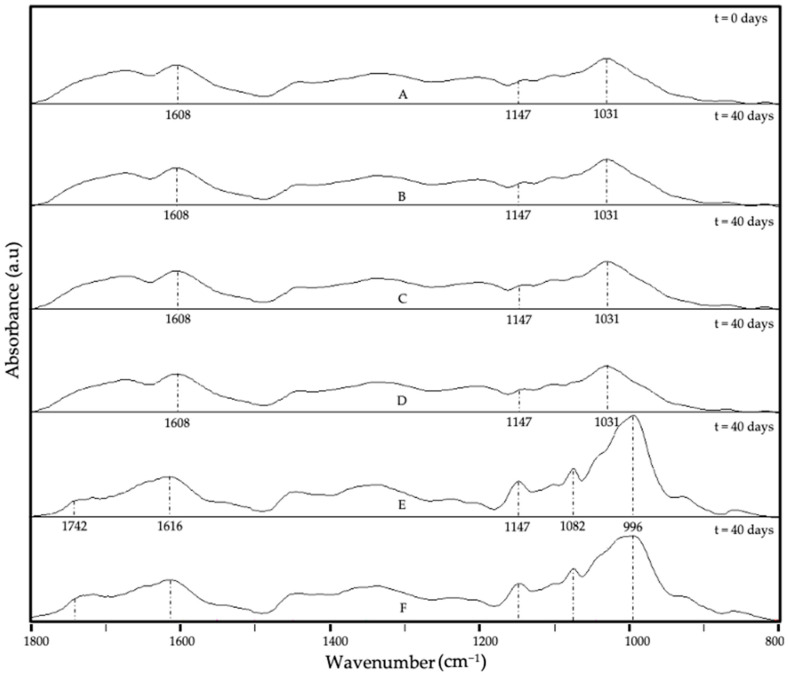
Spectral evolution of attenuated total reflection–Fourier transform infrared spectroscopy (ATR-FTIR) for the EAPG-dried CCX stored under various conditions: (**A**) 0 days; (**B**) 40 days, 0% RH, 4 °C; (**C**) 40 days, 0% RH, 21 °C; (**D**) 40 days, 23% RH, 21 °C; (**E**) 40 days, 56% RH, 21 °C; (**F**) 40 days, UV light exposure. The spectra were maximized to the band with the highest intensity in the wavenumber range between 1800 and 800 cm^−1^.

**Figure 7 foods-13-03280-f007:**
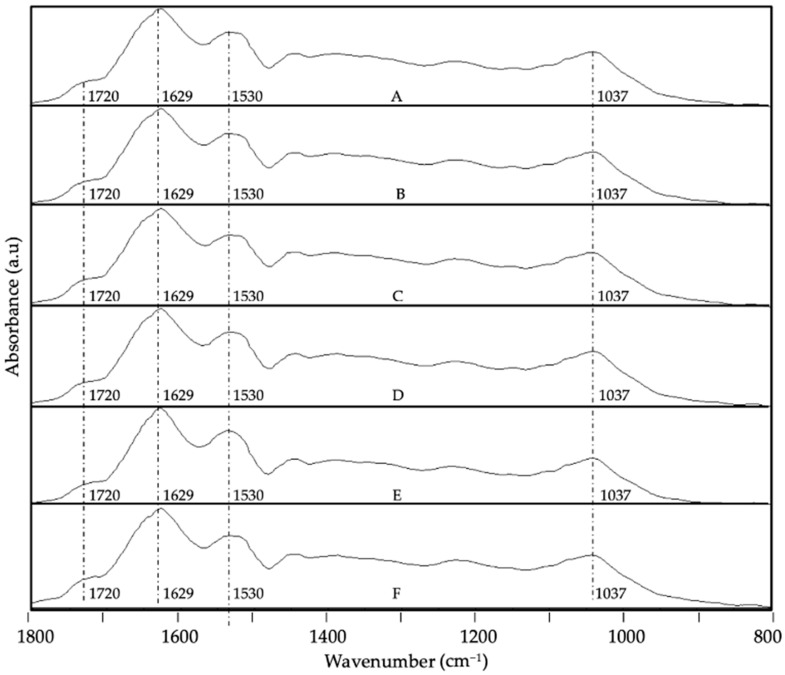
Spectral evolution of attenuated total reflection–Fourier transform infrared spectroscopy (ATR-FTIR) for WPC-CCX 1:1 *w*/*w* EAPG-derived microcapsules stored under various conditions: (**A**) 0 days; (**B**) 40 days, 0% RH, 4 °C; (**C**) 40 days, 0% RH, 21 °C; (**D**) 40 days, 23% RH, 21 °C; (**E**) 40 days, 56% RH, 21 °C; (**F**) 40 days, UV light exposure. The spectra were maximized to the band with the highest intensity in the wavenumber range between 1800 and 800 cm^−1^.

**Figure 8 foods-13-03280-f008:**
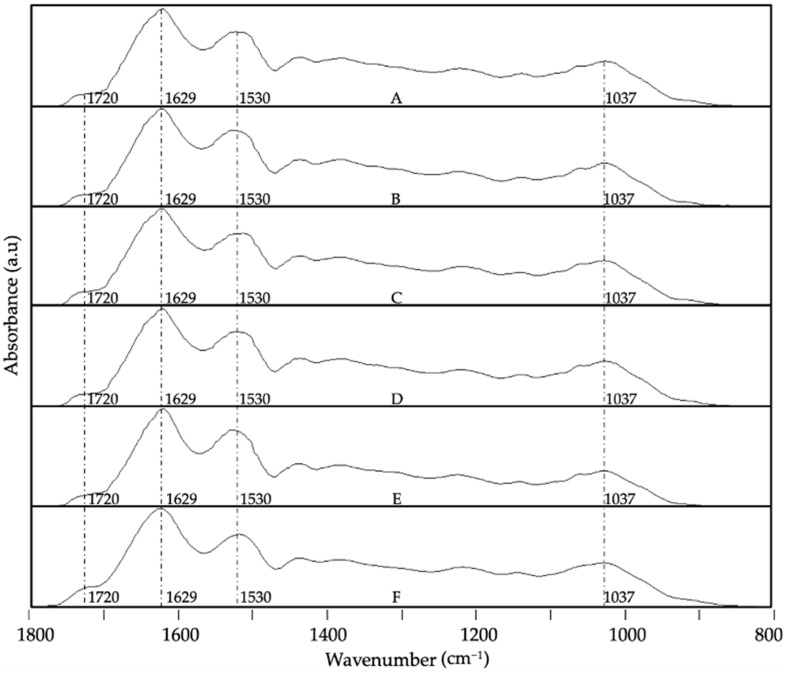
Spectral evolution of attenuated total reflection–Fourier transform infrared spectroscopy (ATR-FTIR) for WPC-CCX 2:1 *w*/*w* EAPG-derived microcapsules stored under various conditions: (**A**) 0 days; (**B**) 40 days, 0% RH, 4 °C; (**C**) 40 days, 0% RH, 21 °C; (**D**) 40 days, 23% RH, 21 °C; (**E**) 40 days, 56% RH, 21 °C; (**F**) 40 days, UV light exposure. The spectra were maximized to the band with the highest intensity in the wavenumber range between 1800 and 800 cm^−1^.

**Figure 9 foods-13-03280-f009:**
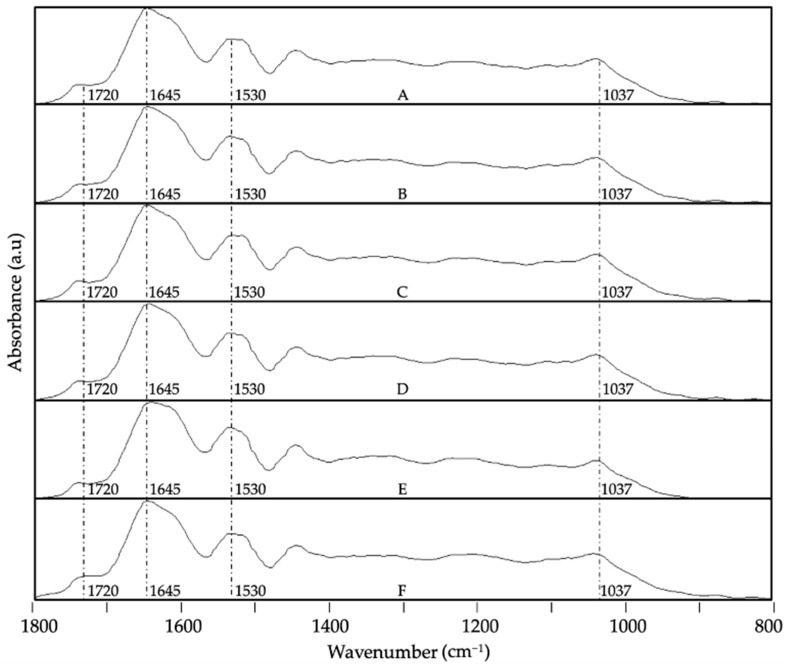
Spectral evolution of attenuated total reflection–Fourier transform infrared spectroscopy (ATR-FTIR) for ZN-CCX 1:1 *w*/*w* EAPG-derived microcapsules stored under various conditions: (**A**) 0 days; (**B**) 40 days, 0% RH, 4 °C; (**C**) 40 days, 0% RH, 21 °C; (**D**) 40 days, 23% RH, 21 °C; (**E**) 40 days, 56% RH, 21 °C; (**F**) 40 days, UV light exposure. The spectra were maximized to the band with the highest intensity in the wavenumber range between 1800 and 800 cm^−1^.

**Figure 10 foods-13-03280-f010:**
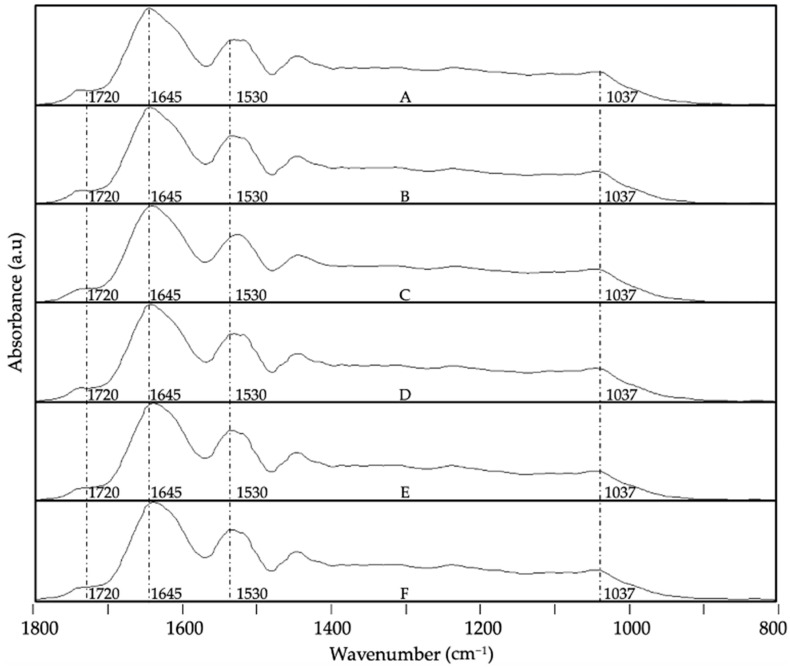
Spectral evolution of attenuated total reflection–Fourier transform infrared spectroscopy (ATR-FTIR) for ZN-CCX 2:1 *w*/*w* EAPG-derived microcapsules stored under various conditions: (**A**) 0 days; (**B**) 40 days, 0% RH, 4 °C; (**C**) 40 days, 0% RH, 21 °C; (**D**) 40 days, 23% RH, 21 °C; (**E**) 40 days, 56% RH, 21 °C; (**F**) 40 days, UV light exposure. The spectra were maximized to the band with the highest intensity in the wavenumber range between 1800 and 800 cm^−1^.

**Table 1 foods-13-03280-t001:** Physicochemical properties of CCX dried via EAPG process and encapsulated with WPC and ZN.

Sample	Particle Size (μm)	Moisture (%)	Color			
			L*	a*	b*	ΔE*
CCX-EAPG	10.01 ± 1.84 ^a^	4.67 ± 1.32 ^a^	37.56	5.11	7.50	
WPC-CCX 1:1	6.74 ± 2.57 ^ab^	5.17 ± 1.93 ^a^	39.14	4.21	8.96	2.33
WPC-CCX 2:1	7.24 ± 2.49 ^ab^	4.78 ± 0.97 ^a^	39.46	3.74	8.36	2.50
ZN-CCX 1:1	6.24 ± 1.72 ^b^	4.15 ± 1.31 ^a^	37.79	4.15	7.15	1.05
ZN-CCX 2:1	5.85 ± 1.45 ^b^	4.07 ± 1.04 ^a^	44.06	2.41	9.04	7.21

Mean values ± standard deviation with different superscripts in the same column are significantly different (Tukey test, *p* ≤ 0.05). Color standard deviation ≤ 0.10.

**Table 2 foods-13-03280-t002:** Total soluble polyphenol content and antioxidant activity for CCX dried by EAPG (CCX-EAPG) and encapsulated CCX into whey protein concentrate (WPC) and zein (ZN).

Sample	DPPH Inhibition (%)	TSPs (mg GAE/g Dried CCX)
CCX	89.06 ± 0.02 ^a^	1.13 ± 0.05 ^a^
CCX-EAPG	94.03 ± 0.02 ^b^	1.14 ± 0.07 ^a^
WPC-CCX 1:1	91.60 ± 0.06 ^ab^	1.15 ± 0.04 ^a^
WPC-CCX 2:1	94.07 ± 0.04 ^b^	1.11 ± 0.05 ^a^
ZN-CCX 1:1	85.32 ± 0.02 ^a^	1.21 ± 0.06 ^a^
ZN-CCX 2:1	93.22 ± 0.04 ^b^	1.15 ± 0.05 ^a^

Mean values ± standard deviation with different superscripts in the same column are significantly different (Tukey test, *p* ≤ 0.05). DPPH inhibition percentage of the DPPH reactive. TSPs—total soluble polyphenols. GAE—gallic acid equivalent.

## Data Availability

The original contributions presented in the study are included in the article, further inquiries can be directed to the corresponding author.

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
