# Peer review of "Comparison of the Stability of a Camu Camu Extract Dried and Encapsulated by Means of High-Throughput Electrospraying Assisted by Pressurized Gas"

_foods, 2024, doi:10.3390/foods13203280_

Round 1
Reviewer 1 Report
Comments and Suggestions for Authors
This manuscript investigated the stability of a camu-camu extract dried and encapsulated by means of high throughput electrospraying assisted by pressurized gas. There are some major questions needed to answer:
1. Introduction: the necessity of camu-camu extract encapsulation should be emphasized.
2. Materials and Methods: encapsulation efficiency and loading capacity are very important parameters to evaluate the encapsulation. And these are all missed.
3. Materials and Methods: the uniformity evaluation of encapsulated powder should be presented.
4. Materials and Methods: The DPPH method is very common method. The method description should be shorten and brief.
5. The format of some units are needed to be corrected. Such as cm-1.
6. The particle size is not nano-particle. What is the advantage of this encapsulation technique?
7. 3.3.1: the results analysis are not enough. I suggested the relationship between the TSP with time could convert to the kinetic equation, then compare the degradation K to compare the loss during different storage. The same thing for 3.3.2.
8. 3.4. ATR-FTIR analysis: there are too many figures and very less results and discussion. I suggested to have a table which can summarize the changes of ATR-FTIR data. Otherwise, it is very unclear.
9. It is not enough to show the changes of microcapsule using the ATR-FTIR analysis. The most obvious way is by the changes of particle size and intact of capsule during storage.
10. Also the authors should show the physical properties of capsule such as the solubility. This is very important property during capsule application.
Comments on the Quality of English Language
I am not qualified to assess the quality of English in this paper.
Author Response
- Introduction: the necessity of camu-camu extract encapsulation should be emphasized.
We have revised the manuscript to further emphasize the vulnerability of bioactive compounds present in camu camu, particularly under conditions such as processing and storage.
This aspect has been reinforced in the Introduction section (Lines 61-65).
- Materials and Methods: encapsulation efficiency and loading capacity are very important parameters to evaluate the encapsulation. And these are all missed.
The requested information has been included in section 2.5.
- Materials and Methods: the uniformity evaluation of encapsulated powder should be presented.
The requested information has been added to the revised manuscript. You can find this information in Section 3.1.
- Materials and Methods: The DPPH method is very common method. The method description should be shorten and brief.
We have simplified the description of the DPPH method in the revised version, removing unnecessary details and referring to previous studies that detail this methodology.
- The format of some units are needed to be corrected. Such as cm-1.
The format of units has been revised and corrected.
- The particle size is not nano-pWhat is the advantage of this encapsulation technique?
The main advantage of this technique is that thanks to the use of voltage, particles are dried at room temperature; therefore, this technique is suitable for thermolabile bioactive compounds. The product is obtained as a free-flowing powder. Moreover, it gathers the advantages of the electrospray process, such as high encapsulation efficiency, control of particle size and morphology, and it is available at industrial scale.
This aspect has been clarified in the manuscript's introduction (Lines 97-103).
- 3.1: the results analysis are not enough. I suggested the relationship between the TSP with time could convert to the kinetic equation, then compare the degradation K to compare the loss during different storage. The same thing for 3.3.2.
Thank you for your input on the analysis in sections 3.3.1 and 3.3.2. Our aim was to provide a quick assessment rather than a detailed kinetic analysis. For future studies, we intend to develop a model to compare degradation rates under the studied storage conditions.
- 4. ATR-FTIR analysis: there are too many figures and very less results and discussions. I suggested to have a table which can summarize the changes of ATR-FTIR data. Otherwise, it is very unclear.
In this study, TSP and DPPH were used to quantify the stability of the bioactive, whereas ATR-FTIR was used to provide a qualitative description of the chemical changes that occurred under the studied storing conditions. This aspect has been clarified in section 3.3.
- It is not enough to show the changes of microcapsule using the ATR-FTIR analysis. The most obvious way is by the changes of particle size and intact of capsule during storage.
No significant differences were found in morphology or particle size after 40 days of storage as shown in the following figure. This aspect has been clarified in section 3.3.
|
Sample |
Morphology |
|
|
0 days |
40 days |
|
|
WPC - CCX 1:1 |
|
|
|
WPC - CCX 2:1 |
|
|
|
ZN - CCX 1:1 |
|
|
|
ZN - CCX 2:1 |
|
|
- Also the authors should show the physical properties of capsule such as the solubility. This is very important property during capsule application.
The obtained capsules dissolve completely in water or ethanol, depending on the nature of the encapsulant matrix, since this encapsulation process does not involve any thermal treatment or chemical reaction. This information has been included in section 3.1.

Reviewer 2 Report
Comments and Suggestions for Authors
Dear authors,
Your manuscript is very interesting and aims to promote the application of Electrospraying assisted by pressurized gas as an advanced technique for the production of microcapsules (in this case protein based microcapsules containing camu camu bioactive extract). In addition, you analysed the morphology and storage stability of obtained microcapsules. Manuscript is well organized, but there are some questions and suggestions that should be answered before any further Manuscript processing. My comments are listed below.
1. For the characterization of obtained microcapsules you used only SEM analysis (morphology of microcapsules). I suggest to you that you additionally characterize your microcapsules (Physicochemical characterization of your capsules). Results for some physicochemical parameters, can help you to explained behaviour of encapsulates during storage under different humidity and temperature conditions.
2. Total soluble polyphenols content and antioxidant activity of microcapsules were determined in prepared capsule's solutions. For this purpose, various ethanol concentrations were used to prepare dried CCX (50% v/v), WPC-CCX (33% v/v), ZN-CCX (75% v/v) (line 186-187 and 206-207). Explain why you used these different concentrations of ethanol to prepare the capsule's solutions?
3. For evaluation of antioxidant activity, you used only one assay (DPPH assay). The results of one screening antioxidant assay are not sufficient to discuss the influence of storage on antioxidant activity of microcapsules. In order to overcome the limitations of some antioxidant assays (for examples, due to the hydrophobic nature, the DPPH radical show a tendency to interact with lipophilic molecules), and to ensure a good interpretation of the results, it is desirable to combine several antioxidant assays that have different mechanisms of action. If you can, I suggest that you include some more antioxidant tests in your study.
4. How did you monitor the changes of encapsulates during storage, using ATR-FTIR? On presented WPP-CCX (Figures 6 and 7) and ZN-CCX (Figures 8 and 9) spectra, shifts cannot be clearly defined, as you claim (line 457-466). How did you determine the shift and decrease of intensity of some bands on spectra? Did you conduct pre-processing of ATR-FTIR spectra: smoothing, baseline correction, unit vector normalization? In this case, when no clear differences (shifts) can be seen in the spectra, it is necessary to perform pre-processing of the spectra and Principal Component Analysis. PCA analysis should indicate potential clustering or discrimination among samples (Samples A-F, Figure 6,7,8,9) and to identify variables (wavenumbers-loading plot) that contribute to grouping/separation among samples. Only on the basis of these results can you discuss the impact of storage (under different conditions) on microcapsules.
Author Response
- For the characterization of obtained microcapsules you used only SEM analysis (morphology of microcapsules). I suggest to you that you additionally characterize your microcapsules (Physicochemical characterization of your capsules). Results for some physicochemical parameters, can help you to explained behaviour of encapsulates during storage under different humidity and temperature conditions.
Based on your valuable suggestion, we have incorporated the analysis of particle size distribution, moisture content, and color. This information can be located in section 3.1.
- Total soluble polyphenols content and antioxidant activity of microcapsules were determined in prepared capsule's solutions. For this purpose, various ethanol concentrations were used to prepare dried CCX (50% v/v), WPC-CCX (33% v/v), ZN-CCX (75% v/v) (line 186-187 and 206-207). Explain why you used these different concentrations of ethanol to prepare the capsule's solutions?
The ethanol concentrations employed in this study were deliberately chosen to ensure the optimal solubility of each encapsulating matrix (whey protein and zein) as well as the solubility of the CCX. The difference in ethanol concentrations respond to the different nature of these materials. This aspect has been clarified in section 2.9.
- For evaluation of antioxidant activity, you used only one assay (DPPH assay). The results of one screening antioxidant assay are not sufficient to discuss the influence of storage on antioxidant activity of microcapsules. In order to overcome the limitations of some antioxidant assays (for examples, due to the hydrophobic nature, the DPPH radical show a tendency to interact with lipophilic molecules), and to ensure a good interpretation of the results, it is desirable to combine several antioxidant assays that have different mechanisms of action. If you can, I suggest that you include some more antioxidant tests in your study.
The DPPH method was selected due to its acknowledged simplicity, speed, and reliability in evaluating antioxidant capacity. We are grateful for the recommendation to incorporate additional antioxidant tests, and in future studies, we will consider including assays such as ORAC, ABTS, or FRAP to offer a more comprehensive assessment of antioxidant activity. This aspect has been clarified in section 3.2.
- How did you monitor the changes of encapsulates during storage, using ATR-FTIR? On presented WPP-CCX (Figures 6 and 7) and ZN-CCX (Figures 8 and 9) spectra, shifts cannot be clearly defined, as you claim (line 457-466). How did you determine the shift and decrease of intensity of some bands on spectra? Did you conduct pre-processing of ATR-FTIR spectra: smoothing, baseline correction, unit vector normalization? In this case, when no clear differences (shifts) can be seen in the spectra, it is necessary to perform pre-processing of the spectra and Principal Component Analysis. PCA analysis should indicate potential clustering or discrimination among samples (Samples A-F, Figure 6,7,8,9) and to identify variables (wavenumbers-loading plot) that contribute to grouping/separation among samples. Only on the basis of these results can you discuss the impact of storage (under different conditions) on microcapsules.
Thank you for your detailed observations. For comparison purposes, the baseline of the analyzed spectra was adjusted, and then the spectra were maximized to the band with the highest intensity in the wavenumber range between 1800 and 800 cm−1. This aspect has been clarified in sections 2.11 and 3.4.
The aim of this study was to provide a qualitative analysis. We understand the value of using more advanced methods like Principal Component Analysis (PCA) to perform a more in-depth analysis, but it was not our objective at this stage of the research. For future research, we will consider to use PCA to perform a more in-depth study.

Round 2
Reviewer 1 Report
Comments and Suggestions for Authors
No comments
Reviewer 2 Report
Comments and Suggestions for Authors
I have no any additional comments and suggestions.